# The Interaction between Leaf Allelopathy and Symbiosis with Rhizobium of *Ulex europaeus* on Hawaii Island

**DOI:** 10.3390/plants9020226

**Published:** 2020-02-10

**Authors:** Mika Hozawa, Eiji Nawata

**Affiliations:** Laboratory of Tropical Agriculture, Department of Agriculture, Kyoto University, Kitashirakawa Oiwakecho, Sakyo-ku, Kyoto 606-8502, Japan; nawata@kais.kyoto-u.ac.jp

**Keywords:** adaptation, *Bradyrhizobium*, invasive species, strategies

## Abstract

The objective of this study was to assess the magnitudes of the leaf allelopathy of *Ulex europaeus* in two different habitats, and discuss the driver of the differences, including rhizobia. The magnitudes of leaf allelopathy of the samples collected in two different habitats were assessed by comparing the hypocotyl and radicle lengths of the lettuce seeds tested on the samples. One habitat was in and adjacent to an *Acasia koa* forest, while the other was more than 50 m away. *A. koa* is indigenous to Hawaii and known to have a close symbiotic relationship with *Bradyrhizobium* for nitrogen-fixing. Within the past three years, *U. europaeus* has newly invaded both sampling sites, whereas the *A. koa* forest has been there for several decades. The combined result of both hypocotyl and radicle lengths of the lettuce seeds tested on both sites by linear model and multicomparison analyses showed no significant difference. But the radicle lengths of the lettuce seeds tested on *U. europaeus* sampled in and adjacent to the *A. koa* forest were significantly longer than those of the samples more than 50 m away, as measured by *t*-test (*p* = 0.05). This result suggested that the magnitude of the leaf allelopathy of *U. europaeus* depended on the distance of the habitat from the *A. koa* forest.

## 1. Introduction

Gorse, *Ulex europaeus* L., which belongs to Fabaceae, is a common native species in the Atlantic coast of Europe, e.g., in the United Kingdom, France, Spain, or Portugal [1]. It was introduced to European colonies in the 19th century as an ornament or hedgerow. In addition, the seeds were unintentionally carried in the fur of livestock, such as sheep, and spread worldwide. It is listed as one of the world’s worst invasive alien species [2], and has invaded a wide range of habitats and climates, from temperate to tropical regions [3]. It has been causing serious economic and environmental problems in regions such as the American continent, the Hawaiian archipelago, Réunion, Australia, and New Zealand.

Plant legume species are known to have symbiotic relationships with rhizobia, i.e., soil-inhabiting bacteria in the root nodules [4]. Rhizobia fix nitrogen from the atmosphere for the growth of the host plant [5]. Although some previous studies have explained the benefits of the symbiosis of the bacteria, such as the metabolic products or oxygen generated during photosynthesis in the host plant promoting the growth and metabolism of the bacteria, the benefit of the bacteria has not been well elucidated [5]. *U. europaeus*, which belongs to the Fabaceae family, is known to have a symbiotic relationship with the rhizobia *Bradyrhizobium* [6], and the existence of *Bradyrhizobium* in soil is key to the successful invasion of *U. europaeus* [6]. 

In addition, allelopathy is known as another strategy used by invasive species to adapt to new environments [7,8]. Allelopathy is defined as any direct or indirect harmful or beneficial effect by one plant (including microorganisms) on another, through the production of chemical compounds that escape into the environment [9]. As the magnitude of the allelopathy of the leaf litter leachate of *U. europaeus* differs significantly according to the habitat [10], allelopathy has been suggested to be one of the adaptation strategies for *U. europaeus* to compete with other species. Bertin et al. [11] stated that the exudation of allelochemicals is multigenically regulated, but few studies have been conducted to elucidate the allelopathic effects in the rhizosphere on weedy species. It is also mentioned that the root exudates play a major role in the rhizosphere, but the effects from the leachate of the leaves cannot be denied. Considering these previous studies, we focused on the allelopathic effect of the leaf litter leachate of one of the weedy species, *U. europaeus*; specifically, on its activities in the rhizosphere.

In the southeast flank of Mauna Kea on Hawaii Island, *U. europaeus* bushes are often seen adjacent to *Acacia koa* forests. *A. koa*, indigenous to Hawaii and belonging to the Fabaceae family, is known to have a symbiotic relationship with *Bradyrhizobium* [6]. After the invasion of *U. europaeus* in the *A. koa* forest, the forest has been greatly diminished, and the lone *A. koa* tree left in the epicenter of *U. europaeus* bushes seems to have been weakened and disappeared in the future (personal observation). In brief, our hypothesis of the prioritized invasion sequence of *U. europaeus* in relation to *A. koa* is that it weakens the allelopathy against *A. koa* at the beginning to utilize the *Bradyrhizobia* on its root, and starts expelling *A. koa* by strong allelopathy after gaining *Bradyrhizobia* in the nodules that start fixing enough nitrogen for adaptation. Leary et al. [6] reported the genetic types of the *Bradyrhizobium* used by *A. koa* in Hakalau Forest on Hawaii Island, which is far from the *U. europaeus*-infested area, and those used by *U. europaeus* in the periphery and epicenter of the infested area. Two same box-PCR fingerprint types of *Bradyrhizobium* used by the two plant species were found, and one common type used by *A. koa* was also used by *U. europaeus*; this was the dominant type in *A. koa*. The chemical structure of nod factor receptors (NFRs) released from the rhizobium enables the rhizobium and the plant species to identify each other; the relationship mutually recognizes the rhizobium and the plant [5]. *U. europaeus* is thought to find the three box-PCR fingerprint types of *Bradyrhizobium* by the unique NFRs at the beginning of the invasion. If the dominant types of *Bradyrhizobia* used by *U. europaeus* are ubiquitous in the soil, *U. europaeus* does not need to rely on the three types of *Bradyrhizobia,* which are common in *A. koa*, to adapt to the new habitats on Hawaii Island. However, it is thought to be a natural process for *U. euroapeus* to start using the three box-PCR fingerprint types of *Bradyrhizobium*, which are known to be used by both *A. koa* and *U. europaeus* at the beginning of the invasion if there is the little dominant type of *Bradyrhizobia* available for *U. europaeus*. The aim of this study was to discuss our hypothesis by comparing the results of the assessment related to the magnitudinal difference in the leaf allelopathy according to the habitats.

## 2. Methods and Materials

### 2.1. Plant Samples

The leaf samples were obtained from seven mother trees on Hawaii Island at coordinates 19°41′13.6″ N and 155°26′44.2″ W (two mother trees were in and two mother trees were adjacent to the *A. koa* forest, altitude: 1888 m) and 19°41′14.0″ N and 155°26′46.2″ W (three mother trees were more than 50 m away from the *A. koa* forest, altitude: 1945m) on March 8, 2018 (Figure 1). In the past three years, both places have been invaded by *U. europaeus* (personal observation) and partially covered by Kikuyu grass (*Pennisetum clandestinum*). The *A. koa* forest is several decades old, and grows on the substrate called “kipuka,” similar to a sandbank made by the lava flows from Mauna Loa eruption in 1843 [12]. *U. europaeus trees* in and adjacent to the kipuka are sparsely distributed in *A. koa* forest. Each mother tree of *U. europaeus* away from the *A. koa* forest is isolated and is growing in the middle of the lava flow of the Mauna Loa eruption [12]. Leaves 5 cm from the tip of the branches were sampled and air-dried, and then kept in sealed plastic bags.

### 2.2. Experiments

The sandwich method [13] was used to assess the magnitude of the allelopathy. Fifty mg of air-dried leaves were sandwiched by two layers (5mL each) of 0.5% agar (Wako Pure Chemical Industries, Ltd.) made with purified water in a plastic cup of 3 cm diameter with a lid; then, five lettuce seeds (cv. Great Lakes) were put on the top of the agar in each cup. Controls were five lettuce seeds put on only 0.5% agar. Lettuce (*Lactuca sativa*, Asteraceae) of cv. Great Lakes is universally used as a test plant because its seeds germinate rapidly and grow uniformly [13]. After incubating at a temperature of 25 °C for 72 h in complete darkness, the length of the hypocotyls and radicles of the lettuce seeds were measured using a digital caliper. Before the experiment, the plastic cups and lids were disinfected by spraying 95% ethanol and air-drying. Ninety seeds (five seeds in each cup) were tested on the agar with leaves from two types of habitats, in and adjacent to the *A. koa* forest, and more than 50m away from the forest (Figure 1). The experiment was conducted from October 16 to 19, 2018, in the laboratory of Kyoto University, Kyoto, Japan.

### 2.3. Statistical Analyses

Paired *t*-tests compared the length of the hypocotyls and radicles of the lettuce seeds. The length of the hypocotyls and radicles of the lettuce seeds tested on the leaves sampled from the mother trees in the two types of habitats by the sandwich method were compared using linear model and multicomparison analyses.

Statistical analyses were performed using the R software, version 3.6.2 [14] and StatPlus (AnalystSoft Inc., Alexandria, VA, USA). 

## 3. Results and Discussion

The radicles and hypocotyls of the lettuce seeds tested on the leaves from the two different habitats were both significantly shorter than those of the control (by *t*-test, *p* < 0.01, *p* < 0.01, respectively). By comparing the combined results of the hypocotyl and radicle lengths of the lettuce seeds tested on two different habitats, no significant difference was found (Figure 2 and Table 1). The hypocotyl lengths of the lettuce seeds tested on the leaves sampled in and adjacent to the *A. koa* forest and those tested on the leaves sampled more than 50m away from the *A. koa* forest were not significantly different (Figure 3), but the radicle lengths of the lettuce seeds tested on the leaves sampled in and adjacent to the *A. koa* forest were significantly longer than those tested on the leaves sampled more than 50 m away (*p* = 0.05, Figure 4).

The results suggest that the leaf allelopathy of *U. europaeus* against other species, maybe including *A. koa*, was strong. Focusing on the relationship of the leaf allelopathy and the distance from the *A. koa* forest, the magnitude of the leaf allelopathy was weaker in the leaves in and adjacent to the *A. koa* forest compared to those from the other sampling site, because the leaf litter leachate normally strongly enhances the growth of radicles [15]. As the key to successful adaptation and propagation of *U. europaeus* greatly depends on the existence of *Bradyrhizobium* [6], the allelopathy of the leaves might be adjusted when it is close to *A. koa*, which has a symbiotic relationship with *Bradyrhizobium.* Specifically, *U. europaeus* of the sampling sites, which are located in the new invasion area, is suggested to lack a common *Bradyrhizobia* type for both species, because it is still in the early stages of adaptation. However, *A. koa* seemed to be a competitive target for *U. europaeus* because the allelopathy of the leaves of *U. europaeus* grown in and adjacent to the *A. koa* forest was suggested to be stronger compared to the control (Figure 2 and Table 1). After obtaining enough bacteria which are common to *A. koa* and *U. europaeus, U. europaeus* may outcompete *A. koa* to obtain sunlight for photosynthesis. As mentioned, three common types of *Bradhyrhizobia* for both species have been reported [6]. *U. europaeus* may rely on the common symbiont of *A. koa* until it obtains its dominant type of Bradyrhizobia in the early stages of adaptation. Given that there was the abundant dominant type of *Bradyrhizobium* for *U. europaeus* except for in the *U. europaeus*-infested area in the study covering all genetic types of Bradyrhizobia of Hawaii Island, the hypothesis may be different. However, study to identify all the genetic types of *Bradyrhozobia* in the soil of Hawaii Island has not been done yet, and it could be even impossible. Furthermore, the magnitude of allelopathy of *U. europaeus* leaves of the mother trees, which have invaded areas in and adjacent to the *A. koa* forest in the past three years, should be tested and compared to the data of this study. If the magnitude of the leaf allelopathy of the leaves of the mother trees older than three years and grown in and adjacent to the *A. koa* forest is significantly stronger than that of the leaves sampled in the same habitat for this study, *U. europaeus* may have proceeded to the next sequence of adaptation in relation to *A. koa.* In addition, the interaction between other types of symbiotic nitrogen fixation, such as the genera of the bacterial group *Frankia* and *Myrica faya,* which are also seriously invasive in Hawaii, should be investigated in terms of invasion strategies to better understand our hypothesis in general.

## 4. Conclusions

To conclude, the results of this study show that the radicle lengths of the lettuce seeds tested on the leaf samples obtained from the mother trees in and adjacent to the *A. koa* forest were significantly shorter than those tested on the samples from more than 50 m away. This may be related to multigenic factors, including allelochemicals exuded from many parts of *U. europaeus.* However, the symbiotic relationship of *U. europaeus* with *Bradyrhizobia* in the new invasion sites is important because it causes a magnitudinal difference in the leaf allelopathy, depending on the habitats.

## Figures and Tables

**Figure 1 plants-09-00226-f001:**
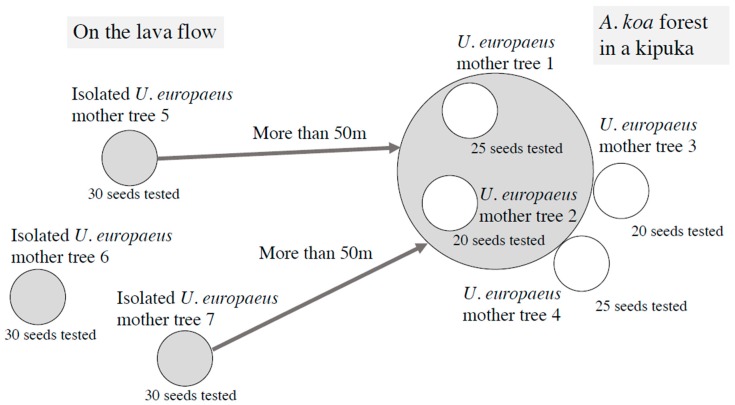
Details of the mother trees used for leaf sampling. *Acasia koa* has formed a forest in the kipuka (sandbank-like, old substrate made by the lava flow). Two mother trees each were in and adjacent to the *A. koa* forest. Three mother trees on the lava were isolated more than 50 m away from the *A. koa* forest. Lettuce seed numbers tested on the leaf samples are indicated below each mother tree.

**Figure 2 plants-09-00226-f002:**
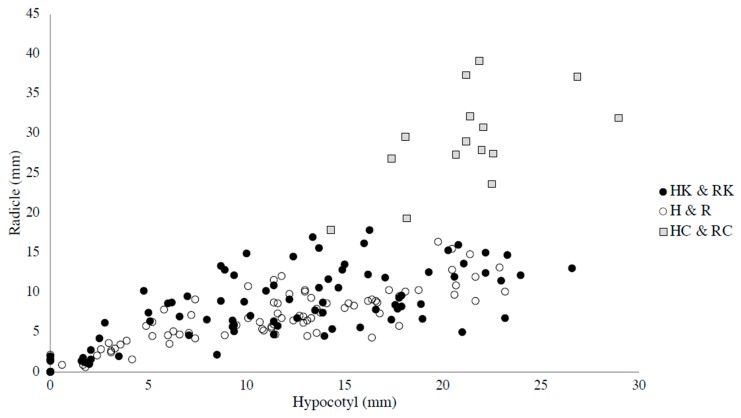
The length of the hypocotyls and radicles of the lettuce seeds tested on the leaves sampled from seven mother trees by the sandwich method were compared using linear model and multicomparison analysws. ●: The hypocotyl and radicle lengths of the lettuce seeds tested on the leaves sampled in and adjacent to the *A. koa* forest. **○**: Hypocotyl and radicle lengths of the lettuce seeds tested on the leaves sampled more than 50 m away from the *A. koa* forest. ◻: Hypocotyl and radicle lengths of the lettuce seeds used as control.

**Figure 3 plants-09-00226-f003:**
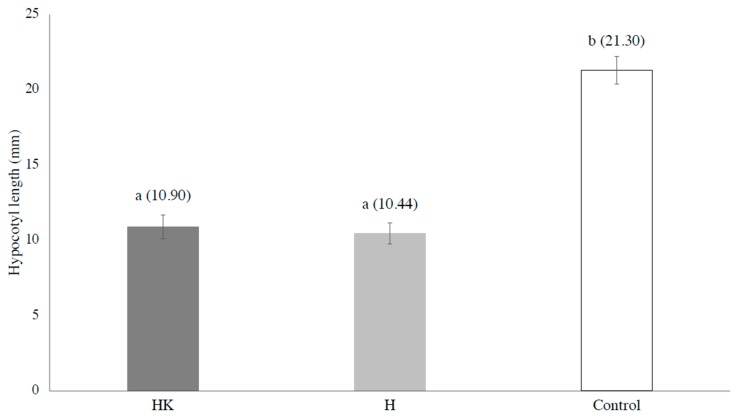
Hypocotyl lengths of lettuce seeds. HK: Hypocotyl lengths of the lettuce seeds tested on the leaves sampled in and adjacent to the *A. koa* forest. H: Hypocotyl lengths of the lettuce seeds tested on the leaves sampled more than 50 m away from the *A. koa* forest. C: Control. The bars represent standard error, and the different letters above bars indicate significant differences (*p* < 0.05). The numbers in the parenthesis are the mean values of the hypocotyls (mm).

**Figure 4 plants-09-00226-f004:**
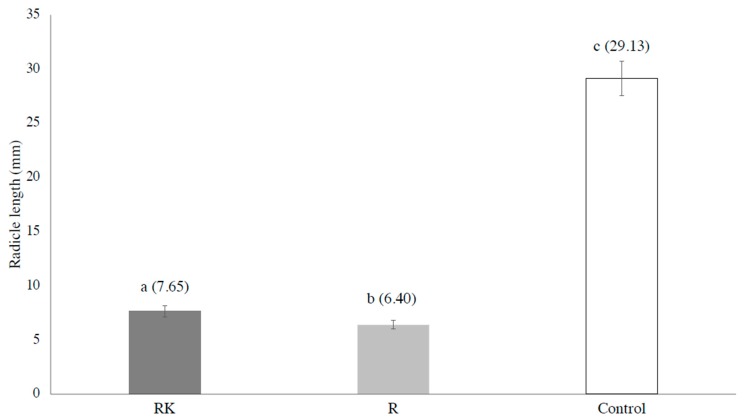
Radicle lengths of lettuce seeds. RK: Radicle of the lettuce seeds tested on the leaves sampled in and adjacent to the *A. koa* forest. R: Radicle of the lettuce seeds tested on the leaves sampled more than 50 m away from the *A. koa* forest. C: Control. The bars represent standard error, and the different letters above bars indicate significant differences (*p* < 0.05). The numbers in the parenthesis are the mean values of the hypocotyls (mm).

**Table 1 plants-09-00226-t001:** The results of the multicomparison of the hypocotyl and radicle lengths of the lettuce seeds tested on the leaf samples. The lengths of the hypocotyls and radicles were significantly shorter than those of the control, but those of the samples tested in and adjacent to the *Acasia koa* forest, and those from more than 50 m away, were not significantly different.

	Estimate	Standard Error	t-Value	*p*-Value
Close to Koa and control	−31.89	2.98	−10.71	<0.001
Away from Koa and control	−33.59	2.98	−11.28	<0.001
Close and away from Koa	−1.7	1.59	−1.07	0.5

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
