# Peer review of "The Interaction between Leaf Allelopathy and Symbiosis with Rhizobium of Ulex europaeus on Hawaii Island"

_plants, 2020, doi:10.3390/plants9020226_

Round 1
Reviewer 1 Report
Paper is innovative. The present study provides evidence that A.absinthiumandP.guajava extract could be proficiently exploited as a botanical herbicide.
Authors characterized potential multigenic properties of allelochemicals exuded from many parts of U. europaeus.
The present study provides evidence, about the invasion strategy of plants.
Reviewer 2 Report
I recommend acceptance of the manuscript for publication in plants in this form
This manuscript is a resubmission of an earlier submission. The following is a list of the peer review reports and author responses from that submission.
Round 1
Reviewer 1 Report
This manuscript describes the influence of the allelopathy produced by Ulex europaeusto Acasia koa. Thank you for your manuscript. This journal is not the best journal for this paperbecause of data quality, data presentation and the discussion. Unfortunately, I regret to inform you that your manuscript cannot be considered for publication in Plant because it does not meet the basic requirements of the journal.
I am very concerned that the authors mentioned that symbiont Bradyrhizobium regulate the magnitude of allelopathy by U. europaeus. However, they do not show any results involved in the contribution of Bradyrhizobium. The conclusion is not convincing and overstatement.
Reviewer 2 Report
The introduction section should have background of the previous chemical and allelopathic effect if found
The abstract should include more details of the results
The values should be inserted on Figure 1 above each peak
The discussion should be more increased with more clear style
the manuscript should include a conclusion section summarized all your results
The English should be revised
Reviewer 3 Report
Allelopathy involves a complex chain of chemical communications between plant species. Hundreds of different compounds released from plants are known to have allelopathic effects on the receiving species. In this short communication, authors clarified the importance of gaining rhizobium for Ulex europaeus to adapt to the new locations by assessing the magnitude of its leaf allelopathy.
In this paper, the allelopathy of the leaves of Ulex europaeus growing adjacent to Acacia koa forest was significantly weaker than that of the leaves growing away from Acacia koa. This result suggested that the magnitude of allelopathy from the leaves of U. europaeus was regulated according to the existence of Bradyrhizobia.
Comments:
The title and aims of the research are coherent to the scope of the journal. The abstract is clearly described and comprehensive. The introduction is properly composed. Material, Methods, and Protocols are very simple. Research is very descriptive, but it can be interesting for new research. The results of the experiments provide novel original data, but the paper doesn't offer new hypothesis. The manuscript is too simple and ignores using relevant methods. The creativity of authors and new innovative views are missing. Arguments need clearer and tighter presentation. The understanding of mechanisms is very limited, as it is restricted to papers that have a particular view and deliberately ignore alternatives, and does not present a balanced view of the evidence. The current manuscript does not adequately describe the novelty of this work.
If it is intended to be a research article, I would have expected more innovative research and critical discussion of the results.